# Microbiological Investigation and Clinical Efficacy of Professional Topical Fluoride Application on *Streptococcus mutans* and *Selemonas sputigena* in Orthodontic Patients: A Randomized Controlled Clinical Trial

**DOI:** 10.3390/microorganisms13112506

**Published:** 2025-10-31

**Authors:** Alessia Pardo, Stefano Marcoccia, Camilla Montagnini, Annarita Signoriello, Elena Messina, Paolo Gaibani, Gloria Burlacchini, Camillo Salgarelli, Caterina Signoretto, Nicoletta Zerman

**Affiliations:** 1Dentistry and Maxillofacial Surgery Unit, Department of Surgery, Dentistry, Pediatrics and Gynecology (DIPSCOMI), University of Verona, Piazzale L.A, Scuro 10, 37134 Verona, Italy; alessia.pardo@univr.it (A.P.); camilla.montagnini@gmail.com (C.M.); nicoletta.zerman@univr.it (N.Z.); 2Dentistry Unit, Department of Surgery, Mater Salutis Hospital, Via Gianella 1, 37045 Verona, Italy; stefano.marcoccia@univr.it (S.M.); camillo.salgarelli@aulss9.veneto.it (C.S.); 3Diagnostic and Public Health Department, University of Verona, 37134 Verona, Italy; paolo.gaibani@univr.it (P.G.); gloria.burlacchini@univr.it (G.B.); caterina.signoretto@univr.it (C.S.); 4Pediatric Dentistry and Oral Hygiene Unit, IRCCS Sacro Cuore-Don Calabria Hospital, 37024 Negrar di Valpolicella, Italy

**Keywords:** dental caries prevention, lactobacilli, oral microbiota, salivary pH, plaque control, fluoride varnish, fluoride gel, randomized controlled trial

## Abstract

Fluoride prophylaxis is a cornerstone in preventing dental caries, a disease for which orthodontic patients are at high risk due to the reduced effectiveness of home oral hygiene and increased plaque accumulation. Recent evidence defines caries as polymicrobial, involving *Streptococcus mutans*, Lactobacilli, and emerging species such as *Selenomonas sputigena.* This prospective, randomized, controlled study evaluated professional topical fluoride in the form of gel and varnish in 68 patients aged 8–17 years wearing fixed orthodontic appliances. Participants were divided into three equal groups: two intervention groups and one control group. Clinical parameters (DMFT, salivary pH, PCR%) and microbiological analyses of plaque and saliva (oral Streptococci, *S. mutans*, *S. sputigena*, Lactobacilli, total bacterial count) were assessed at baseline (T0) and after 4 months (T1), following professional hygiene and fluoride application for the intervention groups. At T1, salivary pH increased in the gel group, and PCR% decreased significantly in all groups, with the most pronounced decrease observed in the varnish group. PCR analysis showed a higher rate of *S. mutans* and *S. sputigena* negativization in intervention groups. Culture-based analyses revealed reductions in oral Streptococci and Lactobacilli in intervention groups, while levels increased in controls. Overall, both clinical and microbiological variables indicated improvements in the fluoride-treated groups compared to controls, highlighting the efficacy of professional fluoride prophylaxis in orthodontic patients.

## 1. Introduction

Orthodontic treatment, which involves the prevention or correction of dental or dento-skeletal malocclusions through the application of fixed or removable appliances, is increasingly utilized among children, adolescents, and adults to enhance both stomatognathic functions and the esthetics of the smile. Despite its undeniable long-term benefits for oral health, this form of treatment—mainly due to its often-extended duration—poses risks to the hard and soft oral tissues of patients. The irregular surface of orthodontic appliances covers a substantial portion of the dental surface, thereby serving as a receptacle for food debris and bacteria, and diminishing the efficacy of both home oral hygiene practices and the self-cleaning action of saliva and oral muscles [1,2,3], despite an increase in salivary flow [1,3]. This leads to an elevated plaque index, resulting in a significant decrease in salivary pH [1,4]. There are both quantitative and qualitative alterations in the oral microbiota [2,5], with a fivefold increase in the total bacterial count [4], accompanied by the proliferation of cariogenic and periodontopathic species [2,6]. The range of ecological changes occurring in the oral cavity of orthodontic patients consequently entails an increased risk of developing gingivitis, periodontitis, white spots, caries, and halitosis [1,2]. Caries is the most prevalent disease globally and represents a public health challenge in all countries, despite being largely preventable [7,8,9]. Indeed, the 2019 Global Burden of Disease identified untreated dental caries on permanent teeth as the most common pathological condition worldwide, and the WHO Global Health Status Report of 2022 estimated that 2 billion people are affected by it [9]. Dental caries is described as a multifactorial disease, resulting from a complex interaction over time among acidogenic bacteria, fermentable carbohydrates introduced through the diet, and host factors including saliva [10], socioeconomic status [11], and fluoride exposure [12]. Significant alterations in the environmental conditions of the oral cavity, such as the consumption of fermentable carbohydrates, promote the colonization of the dental biofilm by cariogenic bacteria. Cariogenic bacterial species, particularly mutans streptococci and Lactobacilli, thus proliferate at the expense of saprophytic species [13,14]. These species are both acidogenic and aciduric: they possess the ability to metabolize fermentable carbohydrates and produce acids, as well as survive in an acidic environment [15]. Fermentable carbohydrates are metabolized through the process of glycolysis, which produces weak acids that lower the salivary pH and cause demineralization of dental tissues, which is the basis of the carious process [16,17]. *Streptococcus mutans* has been extensively recognized as the primary pathogen responsible for dental caries. Nonetheless, the involvement of microorganisms that interact with it, thereby contributing to its functionality and virulence expression through the formation of polymicrobial communities, has garnered significant scholarly attention. The recent study by Cho et al. [15] specifically investigated the role of *Selenomonas sputigena*, a flagellated anaerobe typically identified within the subgingival microbiota of individuals with periodontitis and endodontic infections. The study’s unexpected findings characterized *Selenomonas sputigena* as a pathobiont that collaborates with *Streptococcus mutans*, thereby enhancing its virulence [15]. Notably, *Selenomonas sputigena* alone does not induce carious lesions; however, in a co-infection with *Streptococcus mutans*, it facilitates increased and accelerated acidogenesis, resulting in more extensive carious lesions than those caused by *Streptococcus mutans* alone [15]. Numerous studies indicate that, after the initiation of orthodontic treatment, the levels of *Streptococcus mutans* in the dental biofilm increase significantly [1,2,3,4], as do those of Lactobacilli [2,5]. Specifically, the levels of *Streptococcus mutans* increase a thousandfold within the first month after the placement of a fixed appliance [18].

In the realm of caries prevention, fluoride prophylaxis is recognized as a fundamental strategy and is often regarded as the gold standard [17,19]. This prophylaxis can be administered topically, whereby high concentrations of fluoride are applied directly to exposed tooth surfaces to exert a localized protective effect [20]. Alternatively, fluoride can be administered systemically through the ingestion of specific supplements, such as tablets or drops, or via the consumption of fluoride-fortified foods and beverages, including salt, milk, and water [21,22]. Fluoride exposure can also influence the microbial ecology of orthodontic biofilms by modulating community composition and succession. Such shifts may enhance biofilm resilience by favoring stress-tolerant species and alter acidogenic potential [23], thereby affecting local pH dynamics and the risk of enamel demineralization [24]. Understanding these fluoride-mediated microbial changes provides a mechanistic basis for investigating its role in maintaining oral health during orthodontic treatment. According to the Italian National Guidelines for the promotion of oral health and the prevention of oral diseases in childhood, the post-eruptive preventive effect of fluoride, achieved through topical application, is currently considered more efficacious than the pre-eruptive effect obtained through systemic administration [20,23]. Extensive scientific evidence has documented and reviewed the advantageous effects of topical fluoride as a preventive measure against caries [7,8,21], as well as its role in arresting the progression of incipient carious lesions (white spots) and mitigating their severity [7,8]. The application of topical fluorides results in a significant reduction in the incidence of caries in both primary and permanent dentition [8]. Among the fluoride products designed for topical application, there are toothpastes and mouthwashes intended for home use, mousses and gels that can be self-administered by patients or applied by dental professionals, and varnishes, which necessitate professional intervention [21]. Furthermore, for orthodontic patients, there is the additional option of utilizing adhesive substances for brackets or elastic ligatures that gradually release fluoride over time [21]. We hypothesized that professional topical fluoride prophylaxis, delivered as a gel or a varnish, would reduce cariogenic bacterial load, improve salivary parameters, and enhance plaque control in orthodontic patients. The objective of the present study was to assess the effectiveness of professional topical fluoroprophylaxis in orthodontic patients through the application of a fluoride gel and a fluoride varnish. This evaluation focused on clinical changes in DMFT (Decayed Missing Filled Teeth), salivary pH, PCR% (Plaque Control Record), and changes in the microbiological profile of plaque concerning *Streptococcus mutans* and *Selenomonas sputigena*, as well as variation in saliva with respect to oral Streptococci, Lactobacilli, and total bacterial count.

## 2. Materials and Methods

This study was carried out within the UOSD of Odontostomatology at the Legnago Hospital (VR) in collaboration with the Dentistry and Maxillofacial Surgery Unit of Borgo Roma Hospital and the Institute of Microbiology—Department of Diagnostics and Public Health of University of Verona between March 2024 and August 2025. The care setting in which the recruited patients were treated was outpatient treatment. The study protocol was approved by Ethics Committee of South-East Veneto, Italy with protocol code 55,574 on 9 October 2024 and registered on ClinicalTrials.gov (Identifier: NCT07091890) and conducted in accordance with the CONSORT guidelines for randomized controlled clinical trials [25].

### 2.1. Trial Design

A prospective, randomized, controlled clinical trial was conducted. This study was blinded for the operator who collected the data, the operator who performed the microbiological analysis, and the statistician, in order to ensure a greater validity of the results.

### 2.2. Trial Setting

This study was carried out within the UOSD of Odontostomatology at the Legnago Hospital (VR). The care setting in which the recruited patients were treated was outpatient treatment.

### 2.3. Eligibility Criteria

#### 2.3.1. Inclusion Criteria

1 < DMFT < n-1 (*n* = the number of teeth present in the oral cavity); age between 8 and 17 years; patients wearing fixed orthodontic appliances.

#### 2.3.2. Exclusion Criteria

DMFT = 0 *; professional topical fluoride prophylaxis session carried out in the last 3 months; patients not wearing fixed orthodontic appliances; orthodontic treatment already completed or not started.

* DMFT refers to a caries prevalence index related to the presence of Decayed, Missing, Filled Teeth due to caries (Decayed, Missing, Filled Teeth).

### 2.4. Recruitment and Consent of Participants

All participants of this study were recruited from patients who attended a scheduled orthodontic check-up and met all eligibility criteria. After consent was obtained, the enrolled subject underwent a professional oral hygiene session, and the objectives of this study and the procedures that the patient would undergo were verbally explained in detail to participants and parents/guardians. As this study involved minors, written consent was obtained from both parents. In addition, it was deemed appropriate to consider the assent of the minor participants. Therefore, two different paper-based informed consent forms were provided, i.e., one for the parents and one for the minor patient. More specifically, two different forms were arranged for minors in the 8–11 and 12–17 age groups. The informed consent forms described the purpose of this study, its stages, and the possible risks and benefits of the proposed procedures using a vocabulary suitable for the cognitive abilities and knowledge of the minors and their parents. The investigator was available to answer any questions, and both the participants and their caregivers were given the opportunity to discuss this study with their trusted others and given time to reflect before agreeing to participate. Signing the provided documents represented the formal expression of informed consent, but it was made clear to the patient and their parents (also specified in the informed consent forms) that participation in this study was voluntary and that they could withdraw their consent to participate at any time.

### 2.5. Intervention and Comparator

The patients underwent experimental procedures at T0 and four months later at T1. At T0, all patients had saliva and plaque sampling. Saliva samples were collected by expectorating into a sterile container until 2–3 mL of the sample was obtained. The OptraGate flexible latex-free retractor (Ivoclar Vivadent, Schaan, Liechtenstein) was placed to protect the mucosa from staining and provide better visibility. Plaque was sampled around orthodontic brackets using a probe and transferred onto a Heidemann spatula. The plaque was collected with two micro-brushes placed in tubes containing thioglycolate broth for culture investigation and ethanol for molecular investigation. The samples were labeled with patient identification numbers and stored at 4 °C. Clinical parameters recorded included salivary pH, DMFT, and Plaque Control Record.

Salivary pH was assessed using a pH indicator strip. Salivary pH was measured using MQuant^®^ pH-Indicator Strips (range 0–14; Merck KGaA, Darmstadt, Germany) according to the manufacturer’s instructions. Saliva samples were collected under standardized conditions, preferably on an empty stomach and after rinsing the mouth with water, to minimize the influence of food residues. The strip was directly immersed in the sample, and the resulting color was compared with the reference scale provided by the manufacturer. To ensure measurement reliability, each participant underwent three consecutive measurements, and the mean value was calculated for each individual.

The aim of this assessment was to determine whether the use of fluoride-containing products is associated with an increase in salivary pH. A higher pH creates an environment less favorable to enamel demineralization and may reduce the likelihood of caries development, providing an indirect indication of the protective efficacy of the fluoride products used.

The DMFT index indicates Decayed, Missing, Filled Teeth due to caries. The Plaque Control Record (%) shows the percentage of tooth surface with bacterial plaque, calculated by dividing plaque-covered surfaces by total surfaces examined and multiplying by 100. Four surfaces were examined per tooth: buccal, lingual/palatal, mesial, and distal. The Plaque Control Record was performed last as the plaque-disclosing agent could interfere with saliva pH testing and caries identification. A two-tone plaque disclosing agent (GC, Tokyo, Japan) was applied to all tooth surfaces using pre-soaked pellets until stained. After rinsing, only plaque-bound color remained visible. Patients then underwent professional oral hygiene to remove plaque and tartar using a glycine powder air-polishing device (EMS, Nyon, Switzerland). For calcified deposits, a piezoelectric ultrasonic scaler (EMS, Nyon, Switzerland) or manual scaler (Hu-Friedy, Chicago, IL, USA) was used. Following hygiene treatment, Groups 1 and 2 received topical fluoride prophylaxis. In Group 1, 6–8 g of Lunos® Fluoride Gel (DÜRR DENTAL, Bietigheim-Bissingen, Germany) containing 12,500 ppm fluoride was applied for 4 min using a fitted disposable tray. Patients briefly rinsed with water. In Group 2, Fluor Protector S varnish (Ivoclar Vivadent, Schaan, Liechtenstein) with 7700 ppm fluoride was distributed over dental surfaces using a micro-brush. Patients were instructed not to rinse and avoid eating or drinking for 60 min. The varnish formed a fluoride-rich layer on the enamel, increasing the concentration fourfold. The resin component enabled slow fluoride release, minimizing ingestion. All patients were scheduled for follow-up appointments at T1 (four months later), following identical procedures as T0.

### 2.6. Microbiological Analysis

The microbiological analysis of the saliva and plaque samples was carried out within the “Department of Diagnostics and Public Health, Microbiology Section”, at the University of Verona. The aim of this analysis was to determine the total bacterial count, the presence of oral streptococci and lactobacilli in both the saliva and plaque samples by culture methods, and to detect the presence for *Streptococcus mutans* and *Selenomonas sputigena* in the plaque samples by molecular methods such as end-point PCR. A culture study was performed on the saliva and dental plaque to obtain a semi-quantitative bacterial count, expressed as Colony Forming Unit/mL (CFU/mL). After appropriate dilution, the samples were plated onto different culture media: Brain Heart Infusion (BHI) Agar, a non-selective nutrient medium to assess the total bacterial count (Sigma Aldrich, St. Louis, MO, USA); Mitis Salivarius Agar, selective for oral streptococci (Sigma Aldrich, St. Louis, MO, USA); and Rogosa SL Agar (Sigma Aldrich, St. Louis, MO, USA), selective for lactobacilli. The plates were incubated at 37 °C for 24 h. Dental plaque samples were also used to detect the presence of *S. mutans* and *S. sputigena* DNA by end-point PCR using specific primer pairs [24,26]. Bacterial genomic DNA from the plaque was extracted using the GenElute™ Bacterial Genomic DNA kit (Sigma-Aldrich, St. Louis, MO, USA) according to the manufacturer’s instructions. The extracted DNA was stored at −20 °C until use.

The primer sequences and amplification conditions used for the detection of *Streptococcus mutans* and *Selenomonas sputigena* were specified to ensure methodological transparency and reproducibility. For *Selenomonas sputigena* (amplicon size: 478 bp), the forward primer was 5′-AGAGTTTGATCCTGGCTCAG-3′ and the reverse primer was 5′-CTCAATATTCTCAAGCTCGGTT-3′ [27]. The thermal profile consisted of 35 cycles: 94 °C for 5 min, 94 °C for 1 min, 54 °C for 30 s, 72 °C for 1 min, and a final extension at 72 °C for 5 min. 

For *Streptococcus mutans* (amplicon size: 433 bp), the forward primer was 5′-GGCACCACAACATTGGGAAGCTCAGTT-3′ and the reverse primer was 5′-GGAATGGCCGCTAAGTCAACAGGAT-3′ [28]. The thermal profile consisted of 30 cycles: 94 °C for 5 min, 94 °C for 30 s, 60 °C for 30 s, 72 °C for 30 s, and a final extension at 72 °C for 5 min.

### 2.7. Outcomes

The primary outcome of this study was to evaluate the microbiological effectiveness of professional topical fluoride prophylaxis, delivered as a gel or a varnish, in orthodontic patients using an intention-to-treat (ITT) approach. Microbiological end-points included the following:−The end-point PCR analysis of plaque samples: the detection and negativization of *Streptococcus mutans* and *Selenomonas sputigena*, the key cariogenic and emerging bacterial species.−The culture-based analysis of saliva samples: the quantification of total oral Streptococci and Lactobacilli, as well as total bacterial load, to assess shifts in microbial composition.−Secondary outcomes focused on clinical changes, including the following:−DMFT index (Decayed, Missing, and Filled Teeth): to monitor changes in caries status over the study period.−Salivary pH: measured with a calibrated pH meter to evaluate acid–base changes in the oral environment.−Plaque Control Record (PCR%): calculated to quantify plaque accumulation on tooth surfaces.

All parameters were assessed at baseline (T0) and after 4 months (T1), following professional hygiene and fluoride application for the intervention groups. Improvements in microbial composition and reductions in cariogenic and emerging species were considered primary indicators of the efficacy of fluoride prophylaxis, while clinical parameters served to support and contextualize these findings.

### 2.8. Harms

All adverse events (AEs) associated with the application of a fluoride gel and a fluoride varnish were monitored and recorded throughout this study. AEs were defined as any undesirable effects observed or reported by participants, regardless of their relationship to the intervention. Data on AEs were collected through direct participant interviews and clinical observations during follow-up visits. AEs were classified by severity (mild, moderate, severe) and duration (transient, persistent). All AEs were systematically documented in the study database, including details on the nature, intensity, and resolution of each event.

### 2.9. Sample Size

Sample size calculation was based on data from a study on the topic [29]. For the two fluoride treatment groups, a 25% reduction in the “Plaque Control Record” index was hypothesized compared to the baseline value (thus −13.7). Using the variance of the change in the control group, an estimate of the sample size was obtained using the paired Student’s *t*-test—10 subjects per group—which, considering an estimated 15% drop-out rate, became 12 patients per group (α = 0.05, power = 0.80, and delta = 0.6102).

### 2.10. Randomization

Each participant was randomly assigned to one of three groups: two intervention groups and one control group. The random allocation sequence was generated using Microsoft Excel (version 2309; Microsoft Corporation, Redmond, WA, USA). Assignment was managed by a dentist who was not involved in participant enrollment.

Opaque, sealed envelopes, each containing a coded allocation, were numbered externally and only opened at the time of participant inclusion. This procedure ensured that the investigator responsible for recruiting and treating patients remained blinded to group assignment prior to enrollment.

The study groups were defined as follows:−Group 1 (experimental): professional oral hygiene session (standard of care) plus topical fluoride gel application.−Group 2 (experimental): professional oral hygiene session (standard of care) plus topical fluoride varnish application.−Group 3 (control): professional oral hygiene session (standard of care) only.

### 2.11. Blinding

To further minimize potential bias, the study was double-blinded.

The following operators were blinded to groups’ allocation: (i) the clinician responsible for outcome measurements during follow-up visits; (ii) the technician conducting the microbiological survey; and (iii) the statistician analyzing the data, with no access to group information. Patients were also blinded to the interventions.

Specifically [30], opaque and sealed envelopes, each containing the secret code and bearing on the outside only a number, were opened after patients’ recruitment and informed consent signing so that the investigator involved in the enrollment and treatment could not know in advance which of the 3 treatments to be performed was allocated.

### 2.12. Statistical Methods

The demographic and clinical characteristics of the patients were summarized using descriptive statistical methods. The following parameters were used, i.e., frequency distributions for categorical variables; means and standard deviations for continuous variables with a Gaussian distribution; and median and interquartile range for continuous variables with a non-Gaussian distribution. Deviation from Gaussian distribution was assessed using the Shapiro–Wilk test. Categorical variables were compared using the chi-square test.

For quantitative clinical parameters (plaque index, DMFT, pH) and microbiological variables, comparison between the mean values of the three groups was performed using ANOVA for Gaussian variables and the Kruskal–Wallis test for non-Gaussian variables. To compare means, both for clinical and microbiological variables, at two different time points (T0 and T1) in data with a Gaussian distribution, the paired Student’s *t*-test was used; for non-Gaussian variables, the non-parametric Wilcoxon test for unpaired data was employed. With regard to microbiological variables, the data were analyzed similarly after the logarithmic transformation of the colony count data. 

All analyses were conducted primarily using an intention-to-treat (ITT) approach, including all participants in the groups to which they were originally randomized, regardless of adherence to the intervention or study completion. Missing data were handled using appropriate imputation methods. Per-protocol analyses were performed as a secondary outcome, including only participants who completed the study according to the protocol, to assess the efficacy of the interventions under ideal conditions. The significance level was set at 0.05, and analyses were performed using Stata v.13.0 for Macintosh (StataCorp, College Station, TX, USA). The study design included a control group representing the good clinical practice of oral hygiene without the use of fluoride, which was compared using Dunnett’s test with the two methods of fluoride administration. 

## 3. Results

This experimental study was conducted on a sample of 68 orthodontic patients aged between 8 and 17 years (mean age ± standard deviation = 13.22 ± 1.65), including 24 males (35.3%) and 44 females (64.7%).

The flow of participants throughout this study, including enrollment, randomization, allocation, follow-up, and analysis, is shown in the CONSORT flow diagram (Figure 1).

No adverse events or harms related to the intervention were observed during this study.

No significant differences were found in baseline characteristics (T0) between participants who completed the study and those who dropped out, suggesting limited risk of attrition bias.

### 3.1. Microbiological Results

#### 3.1.1. Results of the Molecular Biology Analysis of Plaque Samples by PCR 

The following results were obtained with the molecular biology analysis by end-point PCR (Polymerase Chain Reaction):Group 1 (22 patients): *S. mutans*: 7 patients were positive at T0 and remained positive at T1; 9 patients were negative at T0 and remained negative at T1; 5 patients were positive at T0 and became negative at T1; 1 patient was negative at T0 and became positive at T1; *S. sputigena*: all patients, except n. 51 and 61, were positive at T0. Among these, 4 patients became negative at T1, while 1 patient became positive at T1. Among the 4 patients who became negative for *S. sputigena*, patient n. 2 also became negative for *S. mutans*.Group 2 (24 patients): *S. mutans*: 9 patients were positive at T0 and remained positive at T1; 9 patients were negative at T0 and remained negative at T1; 3 patients were positive at T0 and became negative at T1; 2 patients were negative at T0 and became positive at T1; *S. sputigena*: All patients were positive at T0 except patient n. 56. Among these, 5 patients became negative at T1.Group 3 (22 patients): *S. mutans*: 5 patients (n. 24, 32, 42, 43, 45) were positive at T0 and remained positive at T1; 2 patients (n. 35, 60) were positive at T0 and became negative at T1; 1 patient (n. 68) became positive at T1; *S. sputigena*: All patients were positive at both T0 and T1.

#### 3.1.2. Results of the Culture-Based Microbiological Analysis on Plaque Samples

Through the culture-based microbiological analysis of plaque samples collected from all patients in each of the three groups at T0 and T1, the following parameters were assessed, i.e., oral Streptococci count, Lactobacilli count, and total bacterial count, expressed in CFU/mL. The findings reveal a statistically significant reduction in the presence of streptococci within the biofilm in the group treated with the fluoride varnish (Table 1). The tables below (Table 1, Table 2 and Table 3) present the results of this analysis, comparing values recorded at T0 with those at T1.

#### 3.1.3. Results of the Culture-Based Microbiological Analysis on Saliva Samples

Through the culture-based microbiological analysis of saliva samples collected from all patients in each of the three groups at T0 and T1, the following parameters were assessed, i.e., oral Streptococci count, Lactobacilli count, and total bacterial count, expressed in CFU/mL. The findings indicate a statistically significant reduction in the presence of streptococci in the saliva of the fluoride varnish group (Table 4). The tables below (Table 4, Table 5 and Table 6) present the results of this analysis, comparing values recorded at T0 with those at T1.

### 3.2. Clinical Results

The table below summarizes the results of the clinical parameters recorded in this study: DMFT, PCR, and pH (Table 7, Table 8 and Table 9). There was a reduction in the Plaque Control Record (PCR), and a statistically significant decrease was observed in both Group 1 and Group 2. Additionally, the pH results indicate a statistically significant reduction for the gel group (1). However, the DMFT index does not exhibit any changes within the individual groups (Table 7).

## 4. Discussion

From a clinical perspective, the DMFT index did not show significant changes during the observation period, an expected result considering the short follow-up, which is insufficient to detect new carious lesions. This finding is consistent with Memarpour et al. [27], who reported significant differences in DMFT only after 12 months in groups treated with the fluoride varnish compared to controls. This result confirms previous findings but also adds novelty by combining clinical and microbiological parameters, providing a more integrated view of fluoride’s preventive effect in orthodontic patients, which has been rarely explored in previous research.

Salivary pH increased in all groups, although to varying degrees. The statistically significant increase observed in Group 1 suggests that the fluoride gel may have a more immediate effect in counteracting salivary acidification [27]. These results are in line with previous studies, such as Mamani-Cori et al., which reported a significant increase in salivary pH after applications of 1.23% and 2% fluoride gels, albeit with a shorter duration of effect. Similarly, an Egyptian study showed a significant increase in salivary pH in patients treated with a fluoride gel [30]. Our study expands this evidence by also evaluating emerging cariogenic species such as *Selenomonas sputigena*, providing new insight into how fluoride may modulate less-studied oral bacteria during orthodontic treatment.

The Plaque Control Record (PCR%), an indicator of plaque accumulation and, indirectly, of home oral hygiene and patient motivation, decreased significantly in all groups. The most pronounced reduction was observed in Group 2, suggesting that fluoride varnishes may exert a protective effect on the tooth surface, hindering bacterial recolonization and plaque adhesion [22]. This finding aligns with Baik et al. [30], who reported significantly lower plaque indices in patients receiving the fluoride varnish compared to controls over 1, 2, 3, and 4 years of follow-up. Professional prophylaxis techniques in fixed orthodontic patients have also been shown to be effective in biofilm removal [29]. Overall, these results confirm the effectiveness of fluoride varnishes in improving plaque control and supporting oral hygiene management. However, absolute PCR% values remained high in all groups, indicating that home hygiene practices alone may be insufficient, likely due to both the technical difficulty of cleaning orthodontic appliances and the reduced compliance and motivation typical of adolescents. Home biofilm management protocols tailored to orthodontic devices are increasingly considered essential [31,32]. Compared with previous studies, our data provide a new perspective by simultaneously analyzing both clinical (PCR%) and microbiological outcomes, allowing a better understanding of how fluoride influences plaque dynamics during orthodontic therapy. These results are also consistent with recent clinical evidence showing that fluoride varnish applications significantly reduce white spot lesions and caries incidence in orthodontic patients [33,34].

From a microbiological perspective, based on molecular analyses (end-point PCR), *Streptococcus mutans* negativity was detected in 41.7% of patients in Group 1 who were positive at baseline T0, in 25.0% of Group 2, and in 28.6% of the control group. These findings substantiate the efficacy of fluoride gel in modulating the prevalence of this cariogenic pathogen [35]. These data are supported by previous clinical studies showing a significant decrease in *S. mutans* colonization after fluoride gel applications, with effects typically lasting 1 to 3 months depending on treatment frequency and concentration [36,37]. In addition, our study provides new evidence on the response of *Selenomonas sputigena*, a recently identified pathobiont associated with increased cariogenicity [15], showing its partial negativization following fluoride treatment. This represents a novel contribution, as the effect of fluoride on this species has not been previously reported in orthodontic populations. Regarding *Selenomonas sputigena*, recently identified as a pathobiont capable of enhancing the cariogenicity potential of *S. mutans* through co-infection mechanisms [15], at T1, a negativization was observed in 20.0% of Group 1 and 22.7% of Group 2, while it was not observed in the control group. Experimental evidence suggests that the modulation of this pathobiont may help reduce plaque acidification and associated caries risk, although direct clinical evidence remains limited, making our finding particularly relevant as one of the first to describe fluoride’s potential role in influencing emerging cariogenic taxa. Notably, dual negativization (*S. mutans* and *S. sputigena*) was observed in only one patient in Group 1, resulting in a significant reduction in caries risk, suggesting that the simultaneous modulation of multiple pathogens may have a relevant clinical impact. Fluoride may exert these effects through the direct inhibition of bacterial metabolic processes, such as membrane transport interference, and indirectly by altering local pH and interspecies interactions within the biofilm, thus modulating microbial virulence [38]. Furthermore, our microbiological results support previous randomized controlled data showing that fluoride exposure can alter the composition of the oral microbiome in orthodontic adolescents, decreasing cariogenic taxa prevalence [39].

The culture-based microbiological analysis revealed a significant reduction in oral Streptococci and Lactobacilli in fluoride-treated groups, while both populations increased in the control group. The reduction in Streptococci is particularly relevant in orthodontic patients, where *S. mutans* typically increase at the expense of eubiotic species such as *S. sanguinis* [1,2,5,32,40]. However, culture-based methods cannot accurately distinguish between pathogenic and commensal species [33]. Analyzing plaque and saliva separately, a statistically significant reduction in oral Streptococci was observed in Group 2. These findings are consistent with the prior clinical evidence demonstrating a market decrease in total salivary Streptococci up to one month after fluoride varnish application [34,37], confirmed by recent reviews attributing the effect to enzymatic inhibition and reduced metabolic activity of oral Streptococci induced by fluoride.

With regard to lactobacilli in dental plaque, the observed reduction, more pronounced in Group 1, is consistent with previous studies [41]. It should be noted that, while certain *Lactobacillus* strains are cariogenic [4], others exhibit probiotic potential and contribute to oral microbiota balance [42]; therefore, their role should be interpreted with caution. By combining both culture-based and molecular approaches, our study adds new information on the complementary effects of fluoride gel and varnish on specific bacterial communities during orthodontic treatment. Total bacterial counts in dental plaque showed a significant decrease, in agreement with the well-known effect of fluoride in limiting the microbial load typically induced by orthodontic treatment [2,5]. Conversely, total salivary bacterial counts increased in all groups, with a significant change in the control group. This phenomenon may reflect greater exposure to nutrient-rich salivary flow and food debris, favoring the growth of bacteria not strictly adhering to dental surfaces. Moreover, while fluoride effectively acts on surface-adherent microbial communities, its direct effect on free-floating salivary microflora is limited, explaining the observed increase, particularly in patients without additional fluoride interventions. Overall, our results suggest that both the fluoride gel and varnish have complementary roles, and their combined evaluation contributes new insights into the ecological effects of fluoride in orthodontic patients.

A direct comparison between the two administration methods revealed complementary outcomes. Group 2 (varnish) showed greater efficacy in reducing PCR%, achieving *S. sputigena* negativization, and decreasing streptococcal and lactobacilli counts in both saliva and plaque. In contrast, Group 1 (gel) showed a greater increase in salivary pH, a higher negativization of *S. mutans*, and more marked reductions in plaque-associated Lactobacilli and total bacterial load. Overall, no clinically substantial differences were observed between the two interventions, suggesting that both the fluoride gel and varnish are effective and may be considered complementary strategies for caries prevention in orthodontic patients. These findings clarify the specific contributions of each formulation and highlight the novelty of assessing fluoride effects on both traditional and emerging cariogenic microorganisms. Compared to previous investigations mainly focused on clinical parameters such as caries or white spot lesion prevention [33,34], the present study provides novel insights into the microbiological modulation induced by fluoride during orthodontic treatment, complementing recent findings on oral microbiome dynamics [39]. The strengths of this study include the combined assessment of clinical and microbiological parameters, the use of both culture-based and molecular methods (end-point PCR), the parallel analysis of saliva and plaque, and attention to both *S. mutans* and emerging species such as *S. sputigena*. These features provide highly relevant data for understanding the complex oral ecology in orthodontic patients and for evaluating the impact of fluoride treatments on diverse bacterial communities. These findings clarify the specific contributions of each formulation and highlight the novelty of assessing fluoride effects on both traditional and emerging cariogenic microorganisms. The main limitations of this study are the short follow-up, which precludes the detection of significant changes in the DMFT index, and the use of end-point PCR, which provides qualitative rather than precise quantitative data on microbial load. Future studies employing qPCR and longer observation periods are warranted to confirm these findings, further clarify potential differences in the preventive efficacy of the fluoride gel versus varnish, and provide more precise guidance for clinical practice in orthodontics. Furthermore, all participants were recruited from a single center, potentially reducing generalizability, and some potential confounders such as dietary intake and physical activity were not assessed. Therefore, results should be interpreted with caution and confirmed in larger multicenter trials.

Another limitation is that an intention-to-treat (ITT) analysis could not be fully applied because outcome data were missing for participants who dropped out. However, baseline comparisons indicated no significant differences between completers and drop-outs, reducing the risk of attrition bias.

## 5. Conclusions

The combined use of fluoride gel and varnish represents a complementary preventive strategy in orthodontics, capable of modulating dental biofilm composition, reducing local acidification, and improving salivary parameters. Personalizing the intervention based on individual patient characteristics and compliance may optimize protection against cariogenic pathogens and support oral hygiene during orthodontic treatment. Further studies with larger cohorts and longer follow-up are needed to confirm these findings and to inform evidence-based clinical guidelines.

## Figures and Tables

**Figure 1 microorganisms-13-02506-f001:**
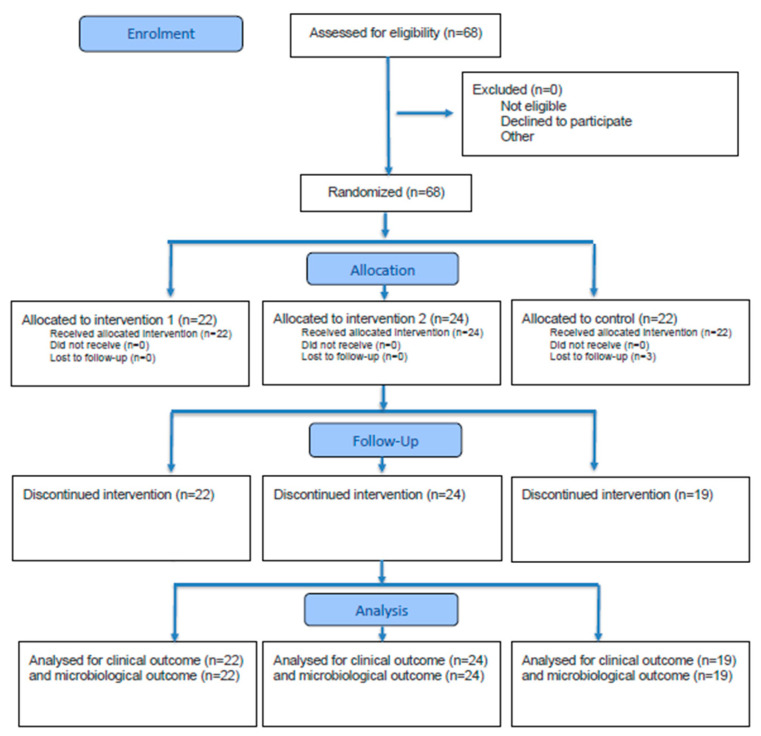
CONSORT flow diagram illustrating participant enrollment, randomization, allocation, follow-up, and analysis.

**Table 1 microorganisms-13-02506-t001:** Groups 1–3: mean ± SD of plaque bacterial counts of oral Streptococci in CFU/mL at T0 and T1, with changes over time.

PLAQUE	Streptococci T0 (Mean ± SD)	Streptococci T1 (Mean ± SD)	Δ Streptococci T1–T0	*p* Value
Total sample (*n* = 68)	1.45 × 10^7^ ± 3.40 × 10^7^	1.45 × 10^7^ ± 3.40 × 10^7^	1.45 × 10^7^ ± 3.40 × 10^7^	0.03 *
Group 1 (gel, *n* = 22)	2.54 × 10^7^ ± 4.28 × 10^7^	2.54 × 10^7^ ± 4.28 × 10^7^	2.54 × 10^7^ ± 4.28 × 10^7^	0.13
Group 2 (varnish, *n* = 24)	1.73 × 10^7^ ± 3.78 × 10^7^	1.73 × 10^7^ ± 3.78 × 10^7^	1.73 × 10^7^ ± 3.78 × 10^7^	0.01 *
Group 3 (control, *n* = 22)	9.90 × 10^5^ ± 1.63 × 10^6^	9.90 × 10^5^ ± 1.63 × 10^6^	9.90 × 10^5^ ± 1.63 × 10^6^	0.25
*p* value	0.43	0.04 *	0.07	

T0 = baseline; T1 = 4-month follow-up; Δ T1–T0 = change from baseline. Data are mean ± SD; CFU/mL = colony-forming units per milliliter. Statistical analyses: Shapiro–Wilk for normality; ANOVA or Kruskal–Wallis for between-group comparisons; paired *t*-test or Wilcoxon for T0–T1; log-transform for microbiological data; Dunnett test vs. control; *p* < 0.05. A total of 68 orthodontic patients randomized into the gel (*n* = 22), varnish (*n* = 24), and control (*n* = 22) groups; an intention-to-treat approach. Sample sizes differ slightly across time points due to drop-outs. Streptococci are measured by a selective culture.); * = statistically significant difference between groups or time intervals.

**Table 2 microorganisms-13-02506-t002:** Groups 1–3: mean ± SD of plaque bacterial counts of oral Lactobacilli in CFU/mL at T0 and T1, with changes over time.

PLAQUE	Lactobacilli T0 (Mean ± SD)	Lactobacilli T1 (Mean ± SD)	Δ Lactobacilli T1–T0	*p* Value
Total sample (*n* = 68)	1.56 × 10^6^ ± 1.21 × 10^7^	1.56 × 10^6^ ± 1.21 × 10^7^	1.56 × 10^6^ ± 1.21 × 10^7^	0.9
Group 1 (gel, *n* = 22)	4.56 × 10^6^ ± 2.13 × 10^7^	4.56 × 10^6^ ± 2.13 × 10^7^	4.56 × 10^6^ ± 2.13 × 10^7^	0.34
Group 2 (varnish, *n* = 24)	4.97 × 10^4^ ± 2.04 × 10^5^	4.97 × 10^4^ ± 2.04 × 10^5^	4.97 × 10^4^ ± 2.04 × 10^5^	0.56
Group 3 (control, *n* = 22)	2.20 × 10^5^ ± 5.08 × 10^5^	2.20 × 10^5^ ± 5.08 × 10^5^	2.20 × 10^5^ ± 5.08 × 10^5^	0.3
*p* value	0.45	0.12	0.6	

**Table 3 microorganisms-13-02506-t003:** Groups 1–3: mean ± SD of plaque bacterial counts of oral total bacteria in CFU/mL at T0 and T1, with changes over time.

PLAQUE	Total Count T0 (Mean ± SD)	Total Count T1 (Mean ± SD)	Δ Total Count T1–T0	*p* Value
Total sample (*n* = 68)	3.32 × 10^7^ ± 1.31 × 10^8^	3.32 × 10^7^ ± 1.31 × 10^8^	3.32 × 10^7^ ± 1.31 × 10^8^	0.02 *
Group 1 (gel, *n* = 22)	5.87 × 10^7^ ± 2.07 × 10^8^	5.87 × 10^7^ ± 2.07 × 10^8^	5.87 × 10^7^ ± 2.07 × 10^8^	0.64
Group 2 (varnish, *n* = 24)	1.49 × 10^7^ ± 2.67 × 10^7^	1.49 × 10^7^ ± 2.67 × 10^7^	1.49 × 10^7^ ± 2.67 × 10^7^	0.2
Group 3 (control, *n* = 22)	2.76 × 10^7^ ± 1.01 × 10^8^	2.76 × 10^7^ ± 1.01 × 10^8^	2.76 × 10^7^ ± 1.01 × 10^8^	0.17
*p* value	0.58	0.09	0.001 *	

* = statistically significant difference between groups or time intervals.

**Table 4 microorganisms-13-02506-t004:** Groups 1–3: mean ± SD of saliva bacterial counts of Streptococci in CFU/mL at T0 and T1, with changes over time.

SALIVA	Streptococci T0 (Mean ± SD)	Streptococci T1 (Mean ± SD)	Δ Streptococci T1–T0	*p* Value
Total sample (*n* = 68)	1.46 × 10^7^ ± 3.40 × 10^7^	1.46 × 10^7^ ± 3.40 × 10^7^	1.46 × 10^7^ ± 3.40 × 10^7^	0.47
Group 1 (gel, *n* = 22)	2.54 × 10^7^ ± 4.28 × 10^7^	2.54 × 10^7^ ± 4.28 × 10^7^	2.54 × 10^7^ ± 4.28 × 10^7^	0.25
Group 2 (varnish, *n* = 24)	1.77 × 10^7^ ± 3.77 × 10^7^	1.77 × 10^7^ ± 3.77 × 10^7^	1.77 × 10^7^ ± 3.77 × 10^7^	0.001 *
Group 3 (control, *n* = 22)	9.67 × 10^5^ ± 2.24 × 10^6^	9.67 × 10^5^ ± 2.24 × 10^6^	9.67 × 10^5^ ± 2.24 × 10^6^	0.5
*p* value	0.35	0.6	0.2	

* = statistically significant difference between groups or time intervals.

**Table 5 microorganisms-13-02506-t005:** Groups 1–3: mean ± SD of saliva bacterial counts of Lactobacilli in CFU/mL at T0 and T1, with changes over time. Sample sizes differ slightly across time points due to drop-outs.

SALIVA	Lactobacilli T0 (Mean ± SD)	Lactobacilli T1 (Mean ± SD)	Δ Lactobacilli T1–T0	*p* Value
Total sample (*n* = 68)	1.77 × 10^5^ ± 1.21 × 10^6^	1.77 × 10^5^ ± 1.21 × 10^6^	1.77 × 10^5^ ± 1.21 × 10^6^	0.99
Group 1 (gel, *n* = 22)	4.93 × 10^4^ ± 1.12 × 10^5^	4.93 × 10^4^ ± 1.12 × 10^5^	4.93 × 10^4^ ± 1.12 × 10^5^	0.63
Group 2 (varnish, *n* = 24)	4.27 × 10^5^ ± 2.04 × 10^6^	4.27 × 10^5^ ± 2.04 × 10^6^	4.27 × 10^5^ ± 2.04 × 10^6^	0.71
Group 3 (control, *n* = 22)	3.29 × 10^4^ ± 6.12 × 10^4^	3.29 × 10^4^ ± 6.12 × 10^4^	3.29 × 10^4^ ± 6.12 × 10^4^	0.1
*p* value	0.4	0.52	0.41	

**Table 6 microorganisms-13-02506-t006:** Groups 1–3: mean ± SD of saliva bacterial counts of total bacteria in CFU/mL at T0 and T1, with changes over time.

SALIVA	Total Count T0 (Mean ± SD)	Total Count T1 (Mean ± SD)	Δ Total Count T1–T0	*p* Value
Total sample (*n* = 68)	2.38 × 10^8^ ± 7.43 × 10^8^	2.38 × 10^8^ ± 7.43 × 10^8^	2.38 × 10^8^ ± 7.43 × 10^8^	0.85
Group 1 (gel, *n* = 22)	1.55 × 10^8^ ± 5.70 × 10^8^	1.55 × 10^8^ ± 5.70 × 10^8^	1.55 × 10^8^ ± 5.70 × 10^8^	0.37
Group 2 (varnish, *n* = 24)	1.56 × 10^8^ ± 2.04 × 10^8^	1.56 × 10^8^ ± 2.04 × 10^8^	1.56 × 10^8^ ± 2.04 × 10^8^	0.7
Group 3 (control, *n* = 22)	4.10 × 10^8^ ± 1.16 × 10^9^	4.10 × 10^8^ ± 1.16 × 10^9^	4.10 × 10^8^ ± 1.16 × 10^9^	0.09
*p* value	0.4	0.3	0.18	

**Table 7 microorganisms-13-02506-t007:** Mean ± standard deviation of DMFT in Groups 1, 2, and 3 at T0 and T1, with the corresponding change between time points.

DMFT	DMFT T0 Mean ± SD	DMFT T1 Mean ± SD	Δ DMFT T1–T0	*p* Value
Total sample (*n* = 68)	3 ± 2.30	3.05 ± 2.36	0.05	0.09
Group 1 (gel, *n* = 22)	2.95 ± 2.23	2.95 ± 2.23	0	0.87
Group 2 (varnish, *n* = 24)	2.58 ± 2.14	2.56 ± 2.19	−0.02	0.12
Group 3 (control, *n* = 22)	3.5 ± 2.52	3.74 ± 2.64	0.24	0.23
*p* value	0.45	0.32	0.56	

T0 = baseline; T1 = 4-month follow-up; Δ T1–T0 = change from baseline. Data are mean ± SD. Statistical analyses: Shapiro–Wilk for normality; ANOVA or Kruskal–Wallis for between-group comparisons; paired *t*-test or Wilcoxon for T0–T1; Dunnett test vs. control; *p* < 0.05. A total of 68 orthodontic patients randomized into the gel (*n* = 22), varnish (*n* = 24), and control (*n* = 22) groups; an intention-to-treat approach. Sample sizes differ slightly across time points due to drop-outs. DMFT = Decayed, Missing, Filled Teeth index measured in situ.

**Table 8 microorganisms-13-02506-t008:** Mean ± standard deviation of PCR in Groups 1, 2, and 3 at T0 and T1, with the corresponding change between time points.

PCR	PCR T0 Mean ± SD	PCR T1 Mean ± SD	Δ PCR T1–T0	*p* Value
Total sample (*n* = 68)	96.29 ± 4.99	89.33 ± 6.27	−6.96	0.001 *
Group 1 (gel, *n* = 22)	94.21 ± 6.79	88.13 ± 6.28	−6.08	0.001 *
Group 2 (varnish, *n* = 24)	97.58 ± 3.72	87.27 ± 5.27	−10.31	0.001 *
Group 3 (control, *n* = 22)	96.73 ± 3.48	94.43 ± 5.32	−2.3	0.14
*p* value	0.35	0.02 *	0.001 *	

T0 = baseline; T1 = 4-month follow-up; Δ T1–T0 = change from baseline. Data are mean ± SD. Statistical analyses: Shapiro–Wilk for normality; ANOVA or Kruskal–Wallis for between-group comparisons; paired *t*-test or Wilcoxon for T0–T1; Dunnett test vs. control; *p* < 0.05. A total of 68 orthodontic patients randomized into the gel (*n* = 22), varnish (*n* = 24), and control (*n* = 22) groups; an intention-to-treat approach. Sample sizes differ slightly across time points due to drop-outs. PCR = Plaque Control Record measured in situ. * = statistically significant difference between groups or time intervals.

**Table 9 microorganisms-13-02506-t009:** Mean ± standard deviation of pH in Groups 1, 2, and 3 at T0 and T1, with the corresponding change between time points.

pH	pH T0 Mean ± SD	pH T1 Mean ± SD	Δ pH T1–T0	*p* Value
Total sample (*n* = 68)	6.97 ± 0.61	7.40 ± 0.64	0.43	0.08
Group 1 (gel, *n* = 22)	6.86 ± 0.56	7.58 ± 0.79	0.72	0.03 *
Group 2 (varnish, *n* = 24)	7.05 ± 0.70	7.34 ± 0.75	0.29	0.23
Group 3 (control, *n* = 22)	6.97 ± 0.58	7.17 ± 0.24	0.2	0.46
*p* value	0.07	0.92	0.12	

T0 = baseline; T1 = 4-month follow-up; Δ T1–T0 = change from baseline. Data are mean ± SD. Statistical analyses: Shapiro–Wilk for normality; ANOVA or Kruskal–Wallis for between-group comparisons; paired *t*-test or Wilcoxon for T0–T1; Dunnett test vs. control; *p* < 0.05. A total of 68 orthodontic patients randomized into the gel (*n* = 22), varnish (*n* = 24), and control (*n* = 22) groups; an intention-to-treat approach. Sample sizes differ slightly across time points due to drop-outs. pH = plaque pH measured in situ. * = statistically significant difference between groups or time intervals.

## Data Availability

Data are available from the corresponding authors upon reasonable request.

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
