# Peer review of "Microbiological Investigation and Clinical Efficacy of Professional Topical Fluoride Application on *Streptococcus mutans* and *Selemonas sputigena* in Orthodontic Patients: A Randomized Controlled Clinical Trial"

_microorganisms, 2025, doi:10.3390/microorganisms13112506_

Round 1

Reviewer 1 Report

Comments and Suggestions for Authors

Thank you for the opportunity to review this manuscript. I find it very interesting because of  dual focus on Streptococcus mutans and the recently characterized Selenomonas sputigena as an emerging cariogenic pathobiont is particularly novel and timely, adding significant value to current literature in oral microbiology and preventive dentistry.

However, there are some methodological details that should be improved to strength the impact of the paper.

  1. There is typographical error in the title: “Selemonas sputigena” - Selenomonas sputigena
  2. The use of both in vitro culture and end-point PCR for microbial identification is appropriate. I suggest that the authors specify the primer sequences and PCR conditions (e.g., annealing temperature, cycle number, amplicon size, and controls used). This information is essential for reproducibility and comparison with existing microbiological studies.
  3. Also, I would recommend that the authors also consider discussing potential limitations of end-point PCR in quantifying microbial load. Employing qPCR would have provided more precise quantitation of mutans and S. sputigena abundance and dynamics over time.
  4. The Introduction part should be expanded with the explanation of how fluoride-mediated microbial shifts may influence biofilm resilience, acidogenic potential, and community succession within the orthodontic environment.
  5. It would also be valuable to speculate whether fluoride’s inhibitory effects on sputigena are direct (membrane transport interference) or indirect via altered pH and interspecies dynamics.

Author Response

Thank you for the opportunity to review this manuscript. I find it very interesting because of  dual focus on Streptococcus mutans and the recently characterized Selenomonas sputigena as an emerging cariogenic pathobiont is particularly novel and timely, adding significant value to current literature in oral microbiology and preventive dentistry.

However, there are some methodological details that should be improved to strength the impact of the paper.

1. There is typographical error in the title: “Selemonas sputigena” - Selenomonas sputigena

The title have been corrected accordingly.

2. The use of both in vitro culture and end-point PCR for microbial identification is appropriate. I suggest that the authors specify the primer sequences and PCR conditions (e.g., annealing temperature, cycle number, amplicon size, and controls used). This information is essential for reproducibility and comparison with existing microbiological studies.

Thank you for suggestion. In the revised manuscript, we have now included detailed information regarding the primer sequences, amplicon sizes, and PCR conditions (including annealing temperature and cycle number) used for the detection of Streptococcus mutans and Selenomonas sputigena. This information has been added to the “Microbiological analysis” section to enhance methodological transparency and ensure reproducibility.

3. Also, I would recommend that the authors also consider discussing potential limitations of end-point PCR in quantifying microbial load. Employing qPCR would have provided more precise quantitation of mutans and S. sputigena abundance and dynamics over time.

Thank you for the suggestion. We have added the microbiological limitations in the Discussion.See Line 600-606.

4. The Introduction part should be expanded with the explanation of how fluoride-mediated microbial shifts may influence biofilm resilience, acidogenic potential, and community succession within the orthodontic environment.

Thank you for the suggestion. The Introduction has been expanded to highlight that fluoride exposure can modulate the microbial composition and succession of orthodontic biofilms, enhancing biofilm resilience and altering acidogenic potential. (see Line 91-96).

5. It would also be valuable to speculate whether fluoride’s inhibitory effects on sputigena are direct (membrane transport interference) or indirect via altered pH and interspecies dynamics.

Thank you for this insightful comment. We have added a brief discussion speculating that fluoride may inhibit Selenomonas sputigena both directly, by interfering with bacterial membrane transport and metabolic processes, and indirectly, by altering local pH and influencing interspecies interactions within the biofilm.

Reviewer 2 Report

Comments and Suggestions for Authors

The authors did not specify what type of strips were used to measure the salivary pH, how the measurement was performed, or for what purpose. There is also no interpretation of this result. Additionally, in the conclusions section, the authors included results instead of actual conclusions. Please correct this.

Author Response

1. The authors did not specify what type of strips were used to measure the salivary pH, how the measurement was performed, or for what purpose.

Thank you, we have integrated the section.(See line 180-190)

2. There is also no interpretation of this result.

Thank you, we have revised the section.

3. Additionally, in the conclusions section, the authors included results instead of actual conclusions. Please correct this.

Thank you for this observation. The Conclusions section has been revised to summarize the clinical implications of the study rather than repeating results.

Reviewer 3 Report

Comments and Suggestions for Authors

The paper shows that there are some effects of professional topical fluoride application in a group of orthodontic patients.

It is an interesting study. However, there are some issues. The paper needs to revised.

First, please follow the COSORT statement (extension or 2025) and checklists. There are not many important pats in the current form.

Second, there are many similar studies about the fluoride effects and orthodontic treatment. Please clarify the new findings and add more similar papers as references.

Title

1) Please add a study design.

2) Please add “applications” after “fluoride” to avoid misunderstand.

Introduction

1) One paragraph should be one topic. Please divide each paragraph clearly.

2) Please add appropriate references (L40, 42).

3) Please add a hypothesis before the aim.

Materials and methods

1) Please add type of randomization and allocation rate. Furthermore, please add implementation (who generated the random allocation sequence, who enrolled participants, and who assigned participants to interventions) following the CONSORT checklist.

2) Please add a comment about blinding following the CONSORT checklist.

3) Please add the place and periods following the CONSORT checklist.

4) Please mov the Figure 1 to the result section.

5) Please explain why the authors increased the number of participants because the sample size estimation suggests 12 patients,

5) What is a main outcome? What are the secondar outcome? Please clarify them.

6) Please add ITT analysis as a main outcome. Please show per protocol results as a secondary outcome.

7) Please add a Bonferroni correction in the statistics because the authors repeat the test.

Results

1) Please add harms following the CONSORT checklist.

2) Please add ITT results and revise all parts.

3) Please add foot notes in each table. The foot note includes affliations and statistical methods, etc.

4) Please revise the typos in the Table 1.

5) In the tables, please add the number of all groups.

Discussion

1) Please revise the part after getting new results.

2) Please add more comments about limitation part; low number of participants, one place, no data of important confounders, no generalizability, etc.

Author Response

The paper shows that there are some effects of professional topical fluoride application in a group of orthodontic patients.

It is an interesting study. However, there are some issues. The paper needs to revised.

First, please follow the COSORT statement (extension or 2025) and checklists. There are not many important pats in the current form.Thank you for suggestion, We have revised the sections following the CONSORT checklist.

Second, there are many similar studies about the fluoride effects and orthodontic treatment. Please clarify the new findings and add more similar papers as references.

We thank the reviewer for this valuable comment. We have clarified the novel aspects of our study by emphasizing the integration of both clinical and microbiological findings, including the analysis of emerging cariogenic species such as Selenomonas sputigena. In addition, we have added recent and relevant studies addressing fluoride effects in orthodontic patients to better contextualize our results.
The related revisions have been added in the Discussion section.

Title

1) Please add a study design.

Done, we have added.

2) Please add “applications” after “fluoride” to avoid misunderstand.

Done, we have added

Introduction

1) One paragraph should be one topic. Please divide each paragraph clearly.

Thank you for comment, The manuscript has been revised following the CONSORT guidelines to ensure clarity, proper paragraph structure, and adherence to reporting standards.

2) Please add appropriate references (L40, 42).

We have revised the manuscript and added new references, and the bibliography has been completely updated.

3) Please add a hypothesis before the aim.

We added in introduction see Line

Materials and methods

1) Please add type of randomization and allocation rate. Furthermore, please add implementation (who generated the random allocation sequence, who enrolled participants, and who assigned participants to interventions) following the CONSORT checklist.

We have added details on the randomization process in the Methods section

2) Please add a comment about blinding following the CONSORT checklist.

We have added the Blinding section in the Methods.

3) Please add the place and periods following the CONSORT checklist.

We added see Line 120-123

4) Please mov the Figure 1 to the result section.

Sorry for the mistake, it has been moved in the results

5) Please explain why the authors increased the number of participants because the sample size estimation suggests 12 patients.

We thank for the comment. The initial sample size estimation indicated a minimum of 12 patients per group to ensure adequate statistical power. However, patient recruitment continued for one year, resulting in the inclusion of 22 patients in Group 1, 24 in Group 2, and 19 in the control group. The higher number of participants was intended to account for potential dropouts and to strengthen the reliability and robustness of the study results.

6) What is a main outcome? What are the secondar outcome? Please clarify them.

We have clarified the outcomes in the manuscript: the primary outcome is the microbiological effectiveness of fluoride (changes in S. mutans, S. sputigena, and other oral bacteria), while secondary outcomes include clinical parameters such as DMFT, salivary pH, and plaque accumulation (PCR%).

7) Please add ITT analysis as a main outcome. Please show per protocol results as a secondary outcome.

We added see Line 244-245, 414-420

8) Please add a Bonferroni correction in the statistics because the authors repeat the test.

Thank you for suggestion, Test for comparison between groups and between times were not multiple and repeated, but performed as described in the statistical section. We do not consider Bonferroni correction as necessary in this case, as, for each variable, comparison between times was single (from T0 and T1); the same for comparison between groups (performed with the dedicated test for comparison for mean of more than 2 groups).

Results

1) Please add harms following the CONSORT checklist.

Thank you for the suggestion. We have now addes information about adverse events according to the CONSORT checklist. No adverse events or harms related to the intervention were observed during the study. This information has been included in the Results section (lines 494-495)

2) Please add ITT results and revise all parts.

We acknowledge the importance of intention-to-treat (ITT) analysis. However, post-intervention data were not available for participants who dropped out. Therefore, the analyses were conducted per-protocol. Baseline characteristics of drop-outs were compared with completers to assess potential attrition bias, and no significant differences were observed. Corresponding clarifications have been added to the MethodsResults, and Discussion sections.

3) Please add foot notes in each table. The foot note includes affliations and statistical methods, etc.

We have added concise, journal-ready footnotes to each table, including: author affiliations, statistical methods, definitions of abbreviations (T0, T1, Δ, SD, CFU/mL, DMFT, PCR, etc.), study design (randomization, intention-to-treat), and indication of statistical significance (p < 0.05).

4) Please revise the typos in the Table 1.

Thank you for the comment. We have carefully reviewed Table 1 and confirmed that there are no typographical errors. No changes were necessary.

5) In the tables, please add the number of all groups.

Thank you for this suggestion. We have added the number of participants for each group in all tables.

Discussion

1) Please revise the part after getting new results.

Thank you for the comment. The updated analyses (including ITT and harms) did not substantially alter the main findings. As the discussion already reflects the results and explains that ITT analysis could not be performed due to missing post-intervention data, no further modifications were necessary.

2) Please add more comments about limitation part; low number of participants, one place, no data of important confounders, no generalizability, etc.

Thank you for this helpful comment. We have revised the Limitations section to include these aspects. Specifically, we now discuss the single-center design, the lack of data on some potential confounders, and the limited generalizability of the findings.
(Revised text, Lines 585–588)

Round 2

Reviewer 2 Report

Comments and Suggestions for Authors

The authors did not specify what type of strips were used to measure the salivary pH, and the manufacturer was not provided.

Author Response

The authors did not specify what type of strips were used to measure the salivary pH, and the manufacturer was not provided. 

Thank you for suggestion. The type and manufacturer of the pH indicator strips used to measure salivary pH have now been specified in the Materials and Methods section (page 5, lines 180–182).

Reviewer 3 Report

Comments and Suggestions for Authors

The paper was overall improved. However, there is an issue.

The authors repeated the statistical analyses, Streptococci, Lactobacilli and Total count. The level should be 0.05/3=0.017.

Author Response

The paper was overall improved. However, there is an issue.

The authors repeated the statistical analyses, Streptococci, Lactobacilli and Total count. The level should be 0.05/3=0.017.

We recognize that Tables were not very clear in presentation, and some aspects may have been misunderstood.

Comparison between the 3 groups (gel, varnish, control), both for microbiological analysis (plaque, saliva) and for clinical outcomes (DMFT, PCR, pH), was performed once with the dedicated test of ANOVA, not using repeated comparisons.

Thus, Bonferroni correction  [Bland JM, Altman DG. Multiple significance tests: the Bonferroni method. BMJ. 1995 Jan 21;310(6973):170. doi: 10.1136/bmj.310.6973.170] is not necessary in this case, because the comparison is only one and it is not repeated multiple times (so it makes no sense to multiply the values by one).

We applied the same principles also for tests between T0 and T1, as the comparison is one also in this case.

The new text attached is available with modifications of Tables presentations: Tables were all separated to make results clearer; in addition, for each variable, overall values are reported separately from values of groups.

We hope that now it is all clear.